# Study on the Wetting Mechanism between Hot-Melt Nano Glass Powder and Different Substrates

**DOI:** 10.3390/mi13101683

**Published:** 2022-10-06

**Authors:** Yifang Liu, Junyu Chen, Gaofeng Zheng

**Affiliations:** 1Department of Instrumental and Electrical Engineering, Xiamen University, Xiamen 361102, China; 2Shenzhen Research Institute of Xiamen University, Shenzhen 518000, China

**Keywords:** wettability, hot-melt glass, flow time, coverage thickness, SiO_2_ and Au substrate

## Abstract

The wettability of molten glass powder plays an essential role in the encapsulation of microelectromechanical system (MEMS) devices with glass paste as an intermediate layer. In this study, we first investigated the flow process of nano glass powder melted at a high temperature by simulation in COMSOL. Both the influence of the different viscosity of hot-melt glass on its wettability on SiO_2_ and the comparison of the wettability of hot-melt glass on Au metal lead and SiO_2_ were investigated by simulation. Then, in the experiment, the hot-melt glass flew and spread along the length of the Au electrode because of a good wettability, resulting in little coverage of the hot-melt glass on the Au electrode, with a height of only 500 nm. In order to reduce the wettability of the glass paste on the Au electrode, a SiO_2_ isolation layer was grown on the surface of golden lead by chemical vapor deposition. It successfully reduced the wettability, so the thickness of the hot-melt glass was increased to 1.95 μm. This proved once again that the wettability of hot-melt glass on Au was better.

## 1. Introduction

Good vacuum packaging [1], even special packaging in a bad environment [2], is an important means to ensure the reliability of MEMS devices. In the middle layer packaging process with nano glass powder, the MEMS sensor can be electrically interconnected with the outside world through the external lead wire of the metal electrode, and the cap, substrate and lead wire can be tightly sealed together by using nano glass powder through hot press bonding. Nano glass powder or the glass frit inter-layer packaging has the advantages of a high tolerance to the surface roughness of the bonding interface, suitable for various materials in the MEMS, electrical insulation characteristics to simplify the electrode lead extraction process and patterning without an additional lithography process by using screen printing [3,4,5]. It has been widely used in the packaging of the MEMS pressure switch [6,7], MEMS gyroscope [8] and accelerometer [9]. Many scholars only describe the packaging principle, packaging process and packaging results of nano glass powder, but there is no report on both the mechanism of infiltration and flow process of hot-melt glass on the substrate. 

After nano glass powder is made on the glass substrate with metal lead through screen printing, during the process of high temperature melting, the wettability of molten nano glass powder on metal lead and the SiO_2_ substrate are different due to a different contact angle, surface tension and adhesion work. After cooling and solidification, the adhesion thickness of the glass powder on the metal lead is different from that on the SiO_2_ substrate. If the height of the glass powder inter-layer on the Au metal lead is less than 10 µm [10], the package will fail.

In order to improve the results of the direct packaging of nano glass powder in the MEMS structure with metal leads, the wettability of nano glass powder in a hot-melt state was investigated. Firstly, the whole flow process of hot-melt nano glass liquid on silver substrate from the starting point to the material interface wall was simulated by COMSOL. Then, the wettability of the hot-melt nano glass powder with a different viscosity on the SiO_2_ substrate was analyzed and compared by simulation. The wetting effect of hot-melt glass with the same viscosity on SiO_2_ and Au substrates were also investigated. Finally, it was verified by experiments that the wettability of hot-melt nano glass on Au metal leads was better than that on SiO_2_, which leads to a too small adhesion thickness. By depositing a SiO_2_ isolation layer on the metal leads, the wettability of hot-melt nano glass on a Au metal lead was successfully reduced, so as to improve its adhesion thickness on the Au.

## 2. Simulation Analysis of Wettability

Wettability is the degree of difficulty for a liquid to adhere to a solid when it contacts with a solid. It is usually determined by the contact angle between the solid–liquid interface and the liquid–gas interface θ. When the contact angle is less than 90°, the liquid can wet the solid. When the contact angle is greater than 90°, the liquid is difficult to wet the solid. Zhu Dingyi et al. [11,12] studied the corresponding relationship between liquid surface tension, solid surface tension and the contact angle. Guan C.H. [13] researched the impact of surface roughness on solid–liquid wettability. Li Wei [14] obtained the contact angle between the hot-melt glass and different substrates through experiments, and the better wettability was attained by polishing the surface of the material. In reference [15], the adhesion work was calculated by measuring the contact angle. The viscosity μ of liquid affected the velocity difference of each layer in the flow, which was one of the key factors affecting the fluidity of the liquid. Reference [16] verified that viscosity μ directly affected the fluidity of the hot-melt alloy liquid, and 1/μ was used to characterize the relationship between the wettability and the temperature of the hot-melt alloy. However, the simulations of the wettability of liquids with a different viscosity on the same substrate and liquids with the same viscosity on different substrates have not been reported.

### 2.1. Simulation Model

The hot-melt glass powder was filled into a silicon pit sputtered with a layer of different substrate materials and heated to reflow to fill the whole pit. Assuming that the bottom radius of the hot-melt glass column was 2 mm and the height was 5 mm, the radius of the sphere equal to its volume was 2.47 mm. Taking the bottom radius of the cylindrical container made of the base material as 3 mm, we got the simulation model as shown in Figure 1.

The material properties of nano glass powder at room temperature were indicated in Table 1: density, 2.221g/cm^3^; viscosity, 1000 Pa·s; and surface tension, 2003.4 mN/m. 

According to the relationship between the surface tension and the temperature in Reference [17], the data in Table 2 were preliminarily sorted out and calculated. 

As shown in Figure 2, the relationship between the contact angle θ and the interfacial tension between solid, liquid and gas can be expressed by “Young’s formula”.
(1)$$\gamma_{\mathrm{sg}}=\gamma_{\mathrm{sl}}+\gamma_{\mathrm{lg}}\cos\theta ,$$

$\gamma_{\mathrm{sg}},\gamma_{\mathrm{sl}}{\;\mathrm{and}\;\gamma }_{\mathrm{lg}}\;$ represent solid–gas interfacial tension, solid–liquid interfacial tension and liquid–gas interfacial tension, respectively.

The corresponding relationship between the liquid surface tension, solid surface tension and contact angle [18,19] was expressed by Equation (2):(2)$$\gamma_{\mathrm{sg}}=\frac{\gamma_{\mathrm{lg}}}{2}\times \left(\sqrt{1+\sin^{2}\theta }+\cos\theta \right),$$

According to the data of hot-melted glass in Table 1 and the surface tension data of the substrate material in Table 2, the contact angle formed when the substrate material and the hot-melted glass were infiltrated and could be calculated by Formula (2).

The liquid–gas surface tension $\gamma_{\mathrm{lg}}\;$= 2003.4 mN/m. At the same time, Equation (2) was transformed as follows:(3)$${\left(\sqrt{1+\sin^{2}\theta }+\cos\theta \right)}^{2}=4\times {\left(\frac{\gamma_{\mathrm{sg}}}{\gamma_{\mathrm{lg}}}\right)}^{2},$$
(4)$$1+\cos\theta \cdot \sqrt{1+{\left(\sin\theta \right)}^{2}}=2\times {\left(\frac{\gamma_{\mathrm{sg}}}{\gamma_{\mathrm{lg}}}\right)}^{2},$$

According to Equation (2), when the contact angle is 90°, the solid surface tension is 1416.6 mN/m. Thus, to consider the positive and negative values of cosθ and convert further:(5)$$\sin^{4}\theta =1-\left[1-2\times {\left(\frac{\gamma_{\mathrm{sg}}}{\gamma_{\mathrm{lg}}}\right)}^{2}\right]{\;}^{2}\left(\gamma_{\mathrm{sg}}<1416.6\cap^{\;}\theta >90{}^{\circ}\right),$$
(6)$$\sin^{4}\theta =1-\left[2\times {\left(\frac{\gamma_{\mathrm{sg}}}{\gamma_{\mathrm{lg}}}\right)}^{2}-1\right]{\;}^{2}\left(\gamma_{\mathrm{sg}}>1416.6\cap^{\;}\theta <90{}^{\circ}\right),$$

According to Equations (4) and (5), the contact angles between each substrate and hot-melted glass could be obtained from the data in Table 1 and Table 2.

Adhesion work is the energy released in the process of adhesion. In the process of adhesion, the surface energy of the solid and liquid is lost, and the surface energy of the solid–liquid interface is generated. The calculation formula of the adhesion work was as follows:(7)$$W_{a}=\gamma_{\mathrm{sg}}+\gamma_{\mathrm{lg}}-\gamma_{\mathrm{sl}},$$

Combined with “Young’s formula” (1), we could obtain:(8)$$W_{a}=\gamma_{\mathrm{sg}}+\gamma_{\mathrm{lg}}-\left(\gamma_{\mathrm{sg}}-\gamma_{\mathrm{lg}}\cos\theta \right)=\gamma_{\mathrm{lg}}\left(1+\cos\theta \right),$$

According to Formula (8), the adhesion work between hot-melted glass and different substrates could be obtained. The contact angle and the adhesion work which were calculated are shown in Table 3.

### 2.2. Simulation of Wettability of Hot-Melt Glass with Different Viscosity on SiO_2_ Substrate

By changing the viscosity of hot-melt glass from 500 Pa·s to 1000 Pa·s, the influence of the viscosity of the hot-melt glass on the flow velocity and wettability of the hot-melt glass was studied with the SiO_2_ as a substrate. Taking the yellow light band as the reference point, the relationship between the viscosity of hot-melt glass and the time needed to flow to the junction of the material bottom and the material wall was explored in this paper. On the SiO_2_ substrate, the steady state of hot-melt glass with a different viscosity flowing to the junction is shown in Figure 3, which corresponds to a different flow time.

Therefore, when the viscosity of the hot-melt glass was 1000, 900, 800, 700, 600 and 500 Pa·s, respectively, the time of the hot-melt glass flowing to the specified distance on the SiO_2_ substrate could also be obtained, as shown in Figure 4. It could be seen that the lower the viscosity of the hot-melt glass, the shorter the flow time to the specified distance, the higher the flow speed and the better the wettability. For the same SiO_2_ substrate, the solid surface energy of hot-melt glass with a different viscosity was the same, but the lower the viscosity was, the higher the wettability was.

### 2.3. Comparison of Wettability of Hot-Melt Glass Solution between SiO_2_ and Au Substrates

Then, the viscosity of the hot-melt glass was kept at 1000 Pa·s, the simulation was carried out on the SiO_2_ and Au substrates and the simulation results, as shown in Figure 5, were obtained. It could be clearly seen from Figure 5 that it took 22 s for the hot-melt glass to flow to the junction on the SiO_2_ substrate and 16.5 s on the Au substrate. With the same viscosity, the surface free energy of the liquid was the same. Yet, combined with the parameters in Table 2, the contact angle between the hot-melt glass and the Au substrate was smaller than that of SiO_2_, and the adhesion work and surface tension on the Au substrate were larger, so the wettability was higher and the flow velocity was higher.

## 3. Experimental

The micro pressure switch was packaged with nano glass powder. The hot-melt glass was transparent and the surface morphology was compact and smooth, as shown in Figure 6. However, because the wettability between the hot-melt glass and the Au electrode were stronger than that between the hot-melt glass and the SiO_2_ substrate, the hot-melt glass flowed rapidly along the length direction of the Au electrode lead and spread out rapidly, resulting in little coverage of this part of the hot-melt glass. After measurement, the thickness of the hot-melt glass on the Au electrode lead was only 500 nm, as shown in Figure 6b. This thickness was not enough to form a sealed package during bonding.

The wettability of hot-melt glass to different materials varies greatly [20,21]. From the above simulation and experimental results, it could be seen that the wettability of hot-melt glass on the Au metal lead was good, so the volume of hot-melt glass passing through the Au metal lead decreased sharply. A silicon wafer sputtered when a large area of Au lines was selected and a thin layer of nano glass powder was manually coated on the whole surface and melted at a high temperature. Figure 7b showed that the amount of hot-melt glass on the Au metal leads was very small, and a small part shrank to the metal free area on the silicon wafer. It was proved that the wettability of glass paste on the Au wire was very strong and the adhesion thickness of the glass paste was not as good as that of the silicon or glass. Based on the verification results, it was proposed that a SiO_2_ isolation layer should be formed on the surface of the metal lead by chemical vapor deposition to reduce the wettability of the glass slurry in this area.

The experimental process and results are shown in Figure 8. The SiO_2_ isolation layer successfully reduced the wettability of the hot-melt glass on the Au metal lead, and this part of the hot-melt glass was consistent with that on the glass sheet. The thickness of the hot-melt glass increased from 500 nm to 1.95 µm. It could be seen that there was a significant difference between the thickness of the glass powder on the metal lead covered with a thin layer of SiO_2_ and that on the metal lead not covered with SiO_2_. This proved once again that the wettability of hot-melt glass on a Au substrate was better.

## 4. Conclusions

The wettability of the molten glass powder was studied by simulation and experimentally. The conclusions obtained in this research are summarized as follows:The smaller the viscosity of the hot-melt glass, the smaller the surface energy of the liquid, the greater the wettability and the higher the flow velocity on SiO_2_. When the viscosity of the molten glass slurry decreased from 1000 Pa·s to 500 Pa·s, the time for the hot-melt glass to flow to the specified interface on the silica substrate decreased from 22 s to 15.4 s.The surface tension of Au metal lead was higher than that of SiO_2_, the contact angle between the Au metal lead and the hot-melt nano glass was smaller and the wettability of the Au metal lead was stronger. When the molten glass slurry with the same viscosity of 1000 Pa·s flowed on the silica and gold substrates, the time to flow to the designated interface was 22 s and 16.5 s, respectively.Compared with SiO_2_, Au had a higher adhesion work, a faster spreading speed and a smaller adhesion thickness in a limited time.By depositing a thin layer of SiO_2_ on the Au metal lead, the flattening speed of hot-melt glass could be effectively reduced and the adhesion height of the nano glass powder could be increased from 500 nm to 1.95 µm.

## Figures and Tables

**Figure 1 micromachines-13-01683-f001:**
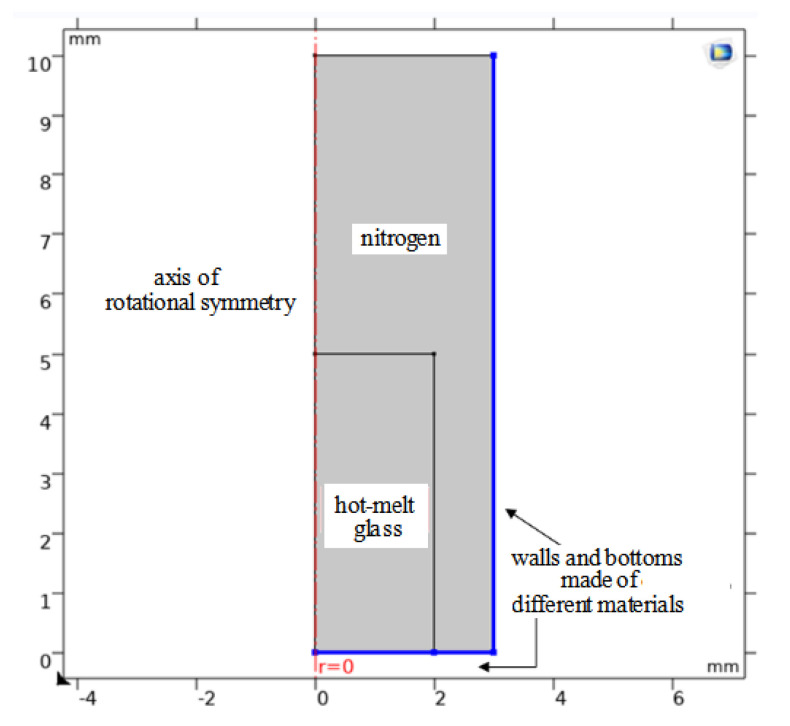
Final simulation model.

**Figure 2 micromachines-13-01683-f002:**
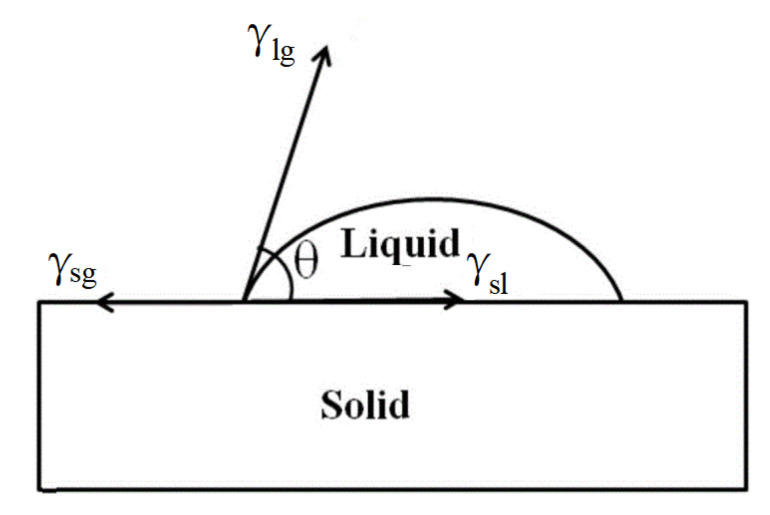
Schematic diagram of surface tension at the junction of contact angle and three phase.

**Figure 3 micromachines-13-01683-f003:**
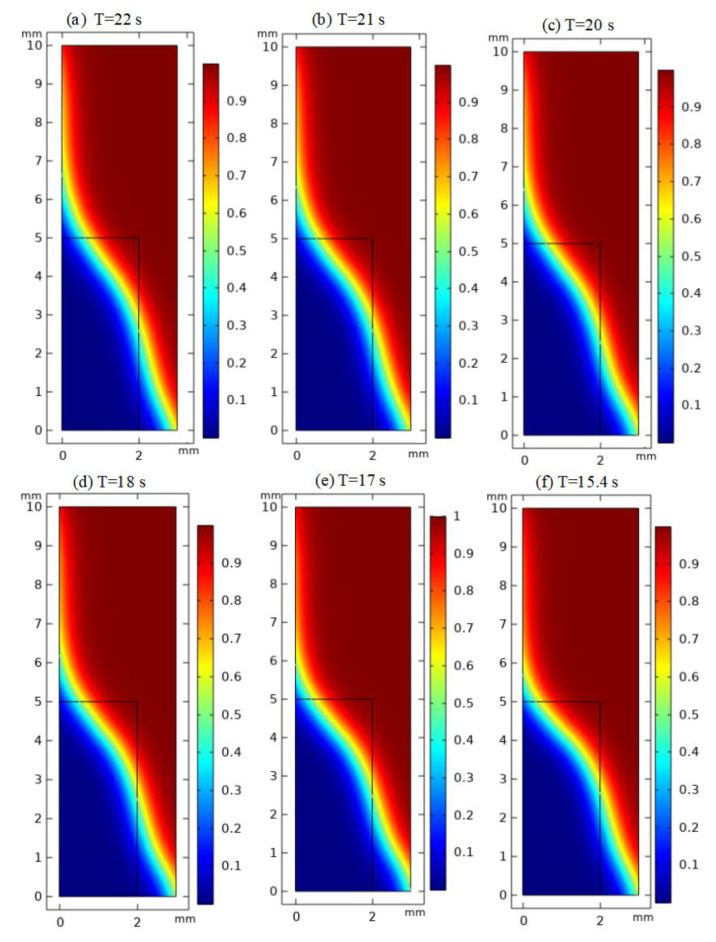
Flow time of hot-melt glass with different viscosity on SiO_2_ substrate: (**a**) the viscosity is 1000 Pa·s; (**b**) the viscosity is 900 Pa·s; (**c**) the viscosity is 800 Pa·s; (**d**) the viscosity is 700 Pa·s; (**e**) the viscosity is 600 Pa·s; (**f**) the viscosity is 500 Pa·s.

**Figure 4 micromachines-13-01683-f004:**
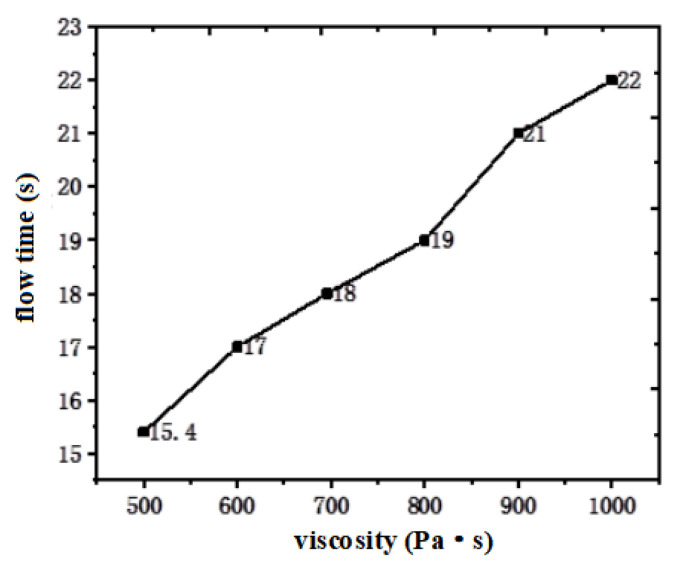
Relationship between flow time and viscosity of hot-melt glass on SiO_2_ substrate.

**Figure 5 micromachines-13-01683-f005:**
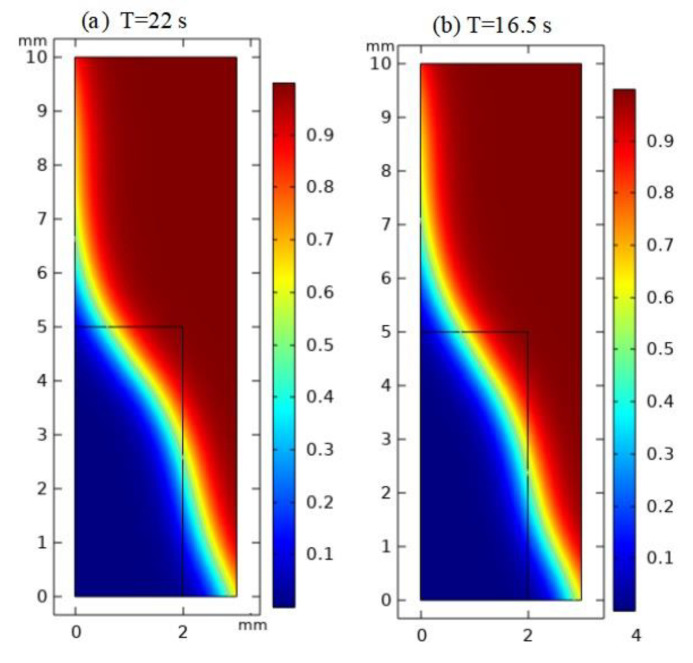
Simulation results of adhesion of hot-melt glass on SiO_2_ and Au substrates: (**a**) SiO_2_; (**b**) Au.

**Figure 6 micromachines-13-01683-f006:**
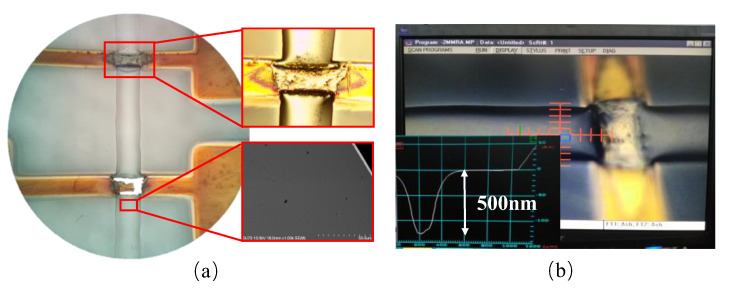
Sintering effect of hot-melt glass: (**a**) slurry morphology; (**b**) measuring diagram of step meter.

**Figure 7 micromachines-13-01683-f007:**
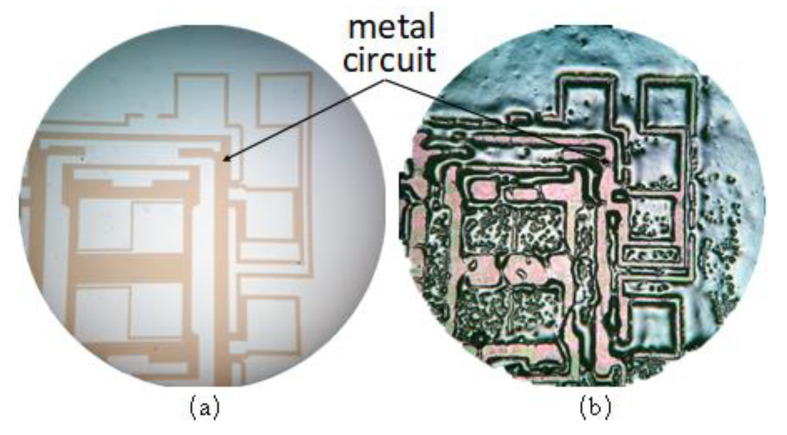
Pre-sintering effect of glass slurry on large-area metal circuit: (**a**) metal circuit; (**b**) morphology of hot-melt glass.

**Figure 8 micromachines-13-01683-f008:**
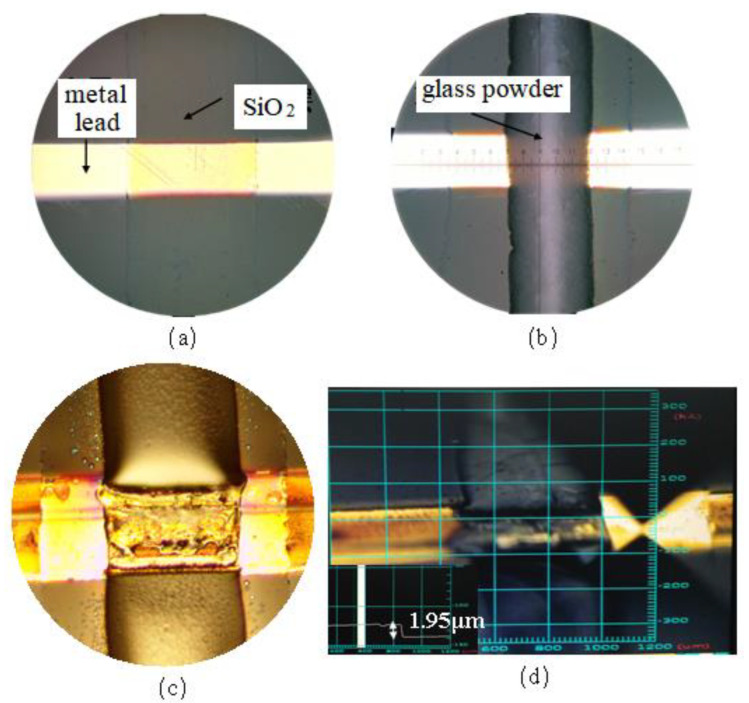
Isolation layer pre-sintering: (**a**) deposition of SiO_2_; (**b**) printing glass paste; (**c**) high temperature hot-melt glass; (**d**) characterization of glass powder thickness.

**Table 1 micromachines-13-01683-t001:** Material properties of nano glass powder at 950 °C.

Density (g/cm^3^)	Viscosity (Pa·s)	Surface Tension (mN/m)
2.221	1000	2003.4

**Table 2 micromachines-13-01683-t002:** Surface tension of silica and gold substrates.

Serial Number	Substrate Material	Density (g/cm^3^)	Melting Point (°C)	Surface Tension at 950 °C (mN/m)
1	SiO_2_	2.2	1723	457.8
2	Au	19.3	1064	1168

**Table 3 micromachines-13-01683-t003:** Contact angle and adhesion work between hot-melt glass and each substrate [14].

Serial Number	Substrate	Surface Tension at 950 °C (mN/m)	Contact Angle (°)	Adhesion Work (mJ/m^2^)
1	SiO_2_	457.8	138.2	509.9
3	Au	1168	103.3	1542.5

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
