# Peer review of "Study on the Wetting Mechanism between Hot-Melt Nano Glass Powder and Different Substrates"

_micromachines, 2022, doi:10.3390/mi13101683_

Round 1

Reviewer 1 Report

This paper investigates the wetting mechanism between hot-melt nano-glass powder and different substrates. Some issues need to be addressed as following:

1. The wettability is defined in the paper, but it seems that the adhesion and fluidity are not very clear. 

2. The silver is used as the substrate in the simulation, but it is not mentioned in the previous content. In addition, the silver substrate is not also used for comparison in the comparison study. Please add some explanations.

3. On Page 5, it mentioned that " the higher the wettability and adhesion were, and the stronger the flattening trend was." what is "the flattening trend"? How can we found "the stronger the flattening trend was"?

4. The simulation results are not validated, how to demonstrate the credibility of the simulation results?

5. There are some typo in the paper. The English writing needs to be polished.

  •  

Author Response

Dear Editors and Reviewers:

Thank you for your letter and for the reviewers’ comments concerning our manuscript entitled “Study on the wetting mechanism between hot-melt nano-glass powder and different substrates” (Manuscript ID: micromachines-1950865). Those comments are all valuable and very helpful for revising and improving our paper, as well as the important guiding significance to our researches. We have studied comments carefully and have made correction which we hope meet with approval. Revised portion are marked in red in the paper. The main corrections in the paper and the responds to the reviewer’s comments are as flowing:

Responds to the reviewer’s comments:

Referee: 1

This paper investigates the wetting mechanism between hot-melt nano-glass powder and different substrates. Some issues need to be addressed as following:

  1. The wettability is defined in the paper, but it seems that the adhesion and fluidity are not very clear. 

When a substance changes from a solid to a liquid, the increase in temperature causes the molecules or atoms to move violently. This makes it impossible for matter to stay in its original position, and flows result. When liquid flows, internal friction occurs between liquid layers due to the different flow velocities of each layer, which hinders the relative motion of liquid layers. The internal friction of liquid flow prevents the relative motion of liquid. This property is called liquid viscosity.

The adhesion work is an important factor to evaluate the adhesion of solid - liquid contact. Adhesion work is the energy released in the process of adhesion. In the process of adhesion, the surface energy of solid and liquid is lost, and the surface energy of solid-liquid interface is generated. The easier it is to infiltrate between solid and liquid, the greater is the value of adhesion work. Under the action of energy, the greater is the energy needed to break away from adhesion theory, that is, the stronger is the adhesion.

In order to match the title, wettability is used to express the interaction between hot-melted glass and substrate.

  1. The silver is used as the substrate in the simulation, but it is not mentioned in the previous content. In addition, the silver substrate is not also used for comparison in the comparison study. Please add some explanations.

It was intended to better compare the wettability of hot-melted glass on different substrates by conducting simulations on a variety of substrates such as silver, gold and silica. However, due to the limitation of experimental conditions and the use of gold as electrode material, silver was not used in the experiment. Therefore, in order to ensure the integrity of the article, this part of the simulation content about silver has been deleted.

  1. On Page 5, it mentioned that " the higher the wettability and adhesion were, and the stronger the flattening trend was." what is "the flattening trend"? How can we found "the stronger the flattening trend was"?

In my mind, stronger flattening trend means faster flow. But after your guidance, I found that this expression is too subjective. We are very sorry for our incorrect writing. So it was deleted from the text.

  1. The simulation results are not validated, how to demonstrate the credibility of the simulation results?

The flow time given by the simulation results can explain the difference of wettability qualitatively.

And in the experiment, the adhesion height of nano glass powder was increased from 500 nm to 1.95 µm by depositing SiO2 thin layer on Au metal lead. This result can be used to explain the credibility of the simulation.

  1. There are some typo in the paper. The English writing needs to be polished.

We are very sorry for our incorrect writing and we have made correction in the paper.

Reviewer 2 Report

In an interesting manuscript entitled "Study on the wetting mechanism between hot-melt nano-glass powder and different substrates," Liu et al. report in detail the results of their simulations and experiments on the flowability, wettability, and adhesion of molten glass powders. What is reported here is important both academically and industrially. It is interesting to see the conclusion that the smaller the viscosity of the hot melt glass, the smaller the surface energy of the liquid, the better the wetting and adhesion, and the greater the flow velocity on SiO2. I would recommend acceptance of this manuscript as it is appropriate for Micromachines readers.

The manuscript would be even better if the following corrections were made.

1) A more detailed explanation of how the data presented in Table 2 was obtained would be appreciated.

2) What surface conditions are assumed for Au, Ag, and SiO2? In general, adhesion depends on the surface condition on the molecular scale. For example, the authors may want to refer to Langmuir 2021, 37, 3982-3995.

Author Response

Dear Editors and Reviewers:

Thank you for your letter and for the reviewers’ comments concerning our manuscript entitled “Study on the wetting mechanism between hot-melt nano-glass powder and different substrates” (Manuscript ID: micromachines-1950865). Those comments are all valuable and very helpful for revising and improving our paper, as well as the important guiding significance to our researches. We have studied comments carefully and have made correction which we hope meet with approval. Revised portion are marked in red in the paper. The main corrections in the paper and the responds to the reviewer’s comments are as flowing:

Responds to the reviewer’s comments:

Referee: 2

In an interesting manuscript entitled "Study on the wetting mechanism between hot-melt nano-glass powder and different substrates," Liu et al. report in detail the results of their simulations and experiments on the flowability, wettability, and adhesion of molten glass powders. What is reported here is important both academically and industrially. It is interesting to see the conclusion that the smaller the viscosity of the hot melt glass, the smaller the surface energy of the liquid, the better the wetting and adhesion, and the greater the flow velocity on SiO2. I would recommend acceptance of this manuscript as it is appropriate for Micromachines readers.

 The manuscript would be even better if the following corrections were made.

  • A more detailed explanation of how the data presented in Table 2 was obtained would be appreciated.

According to the relationship between surface tension and temperature in Reference 17, the data in Table 2 were preliminarily sorted out and calculated.

 Table 2 Surface tension of silica and gold substrates

Serial number

Substrate material

Density(g/cm3)

Melting point ()

Surface tension at 950 ° C(mN/m)

1

SiO2

2.2

1723

457.8

2

Au

19.3

1064

1168

Figure2  Schematic diagram of surface tension at the junction of contact Angle and three phase

As shown in the Figure 2, the relationship between the contact Angle θ and the interfacial tension between solid, liquid and gas can be expressed by "Young's formula".

                          (1)

represent solid-gas interfacial tension, solid-liquid interfacial tension and liquid-gas interfacial tension, respectively.

The corresponding relationship between liquid surface tension, solid surface tension and contact angle [18,19] was expressed by equation (2) :

                  (2)

According to the data of hot-melted glass in Table 1 and the surface tension data of the substrate material in Table 2, the contact Angle formed when the substrate material and hot-melted glass were infiltrated could be calculated by formula (2).

The liquid-gas surface tension = 2003.4 mN/m. At the same time, equation (2) was transformed as follows:

                 (3)

                 (4)

According to Equation (2), when the contact Angle is 90°, the solid surface tension is 1416.6 mN/m. So consider the positive and negative values of  and convert further:  

        (5)

        (6)

According to equations (4) and (5), the contact angles between each substrate and hot melted glass could be obtained from the data in Tables 1 and 2.

Adhesion work is the energy released in the process of adhesion. In the process of adhesion, the surface energy of solid and liquid is lost, and the surface energy of solid-liquid interface is generated. The calculation formula of adhesion work was as follows:

                          (7)

Combined with "Young's formula" (1), we could obtain:

                              (8)

According to formula (8), the adhesion work between hot melted glass and different substrates could be obtained. The contact angle and adhesion work calculated thus were shown in Table 3.

2) What surface conditions are assumed for Au, Ag, and SiO2? In general, adhesion depends on the surface condition on the molecular scale. For example, the authors may want to refer to Langmuir 2021, 37, 3982-3995.

Thank you very much for your guidance! Because the strength of adhesion is related to the contact angle formed between the hot melted glass and different substrates. Therefore, in the simulation process, the surface condition of the substrate is characterized by setting the contact angle between the hot melted glass and the wetting wall.
